# Exploring LOTS in Deep Neural Networks

**Andras Rozsa, Manuel Günther, and Terrance E. Boult**
Vision and Security Technology (VAST) Lab
University of Colorado
Colorado Springs, USA
`{arozsa,mgunther,tboult}@vast.uccs.edu`

## Abstract

Deep neural networks have recently demonstrated excellent performance on various tasks. Despite recent advances, our understanding of these learning models is still incomplete, at least, as their unexpected vulnerability to imperceptibly small, non-random perturbations revealed. The existence of these so-called adversarial examples presents a serious problem of the application of vulnerable machine learning models. In this paper, we introduce the layerwise origin-target synthesis (LOTS) that can serve multiple purposes. First, we can use it as a visualization technique that gives us insights into the function of any intermediate feature layer by showing the notion of a particular input in deep neural networks. Second, our approach can be applied to assess the invariance of the learned features captured at any layer with respect to the class of the particular input. Finally, we can also utilize LOTS as a general way of producing a vast amount of diverse adversarial examples that can be used for training to further improve the robustness of machine learning models and their performance as well.

## 1 Introduction

Due to tremendous progress over the last several years, the most advanced deep neural networks (DNNs) have managed to approach and even surpass human level performance on a wide range of challenging machine learning tasks (Parkhi et al., 2015; Schroff et al., 2015; Szegedy et al., 2015; He et al., 2016). Despite the fact that we are able to design and train learning models that perform well, our understanding of these complex networks is still incomplete. This was highlighted by the discovery of the intriguing properties of machine learning models by Szegedy et al. (2014).

Gaining intuitive insights and, thus, building a better understanding of how these models work has a long history in the literature. For visual recognition tasks, various techniques have been proposed to address the problem of understanding what kind of features are captured and used by learning models (Erhan et al., 2009; Mahendran & Vedaldi, 2016) and how the different internal representations of object classes or input images can be better visualized (Zeiler & Fergus, 2014; Yosinski et al., 2015; Simonyan et al., 2014; Mahendran & Vedaldi, 2016).

While exploring the internal details of DNNs in order to further advance their performance via visualization is particularly relevant and the subject of this paper, recent research has also been focusing on the unpleasant properties revealed by Szegedy et al. (2014). Namely, machine learning models, including the best performing DNNs, suffer from highly unexpected instability as they can confidently misclassify adversarial examples that are formed by adding imperceptible, non-random perturbations to otherwise correctly classified inputs. As DNNs are expected to be robust to small perturbations to their inputs due to their excellent generalization capabilities, the existence of such adversarial perturbations challenges our understanding of DNNs, and questions the utility of such vulnerable learning models in real-world applications. Researchers proposed various techniques to reliably find adversarial perturbations (Szegedy et al., 2014; Goodfellow et al., 2015; Sabour et al., 2016; Rozsa et al., 2016), and demonstrated that adversarial examples can serve a good purpose as well, as they can be successfully used for training to improve both the overall performance and the robustness of machine learning models (Goodfellow et al., 2015; Rozsa et al., 2016).

In this paper, we introduce the layerwise origin-target synthesis (LOTS) that can be efficiently used for multiple purposes. First, it can be applied to visualize the internal representation of an input captured by the learning model at any particular layer. Second, LOTS is capable of forming a vast amount of diverse adversarial examples for each input and we show that such diversity can be beneficial for adversarial training. Finally, derived from the previous possible utilization, LOTS can also be used to explore and assess how stable a particular internal representation of an input is with respect to the class of the input. This paper introduces our novel approach, compares it to related work, and highlights its benefits and possible directions of future work.

## 2 Related Work

While deep neural networks (DNNs) have achieved excellent performance on various tasks, there is an increasing interest to alleviate their vulnerability to adversarial examples. To better understand the sometimes unexpected behavior of learning models, visualizing the learned features is a common practice for gaining intuitive insights. As our work is related to the visualization of feature representations and adversarial instability, this section discusses the relationships to both areas.

### 2.1 Visualization

In order to design and train better performing machine learning models, it is useful to have a clearer understanding about the internal operations and behaviors of these complex models. Otherwise, research aiming to develop more advanced learning models remains restricted to trial-and-error.

Visualization of DNNs was pioneered by Erhan et al. (2009) who displayed high level features learned by various models at the unit level by finding the optimal stimulus in the image space that maximizes the activation of each particular unit. While their approach requires careful initialization, it cannot provide information about the invariance of the inspected units. To address this short-coming, Simonyan et al. (2014) demonstrated how image-specific class saliency maps can be obtained from the last fully connected layers of DNNs, showing areas of the image that are discriminative with respect to a given class. The authors also introduced a technique – image inverting – to generate artificial images for classes that maximize the selected class score. Related to saliency maps, Girshick et al. (2014) showed that identifying regions of images yielding high activations at higher layers can be successfully used for object detection. Zeiler & Fergus (2014) extended visualization to convolutional features. Furthermore, their approach not only finds patches of input images that stimulate a particular feature map, but is also capable of revealing structures within those patches by using a top-down projection.

Although these visualization techniques helped to gain insights into how and why DNNs might work, researchers proposed various approaches to further enhance the produced representations commonly called pre-images. Yosinski et al. (2015) visualized activations on each layer for an image or video, and introduced methods to produce qualitatively clearer and more interpretable visual representations. Similarly, Mahendran & Vedaldi (2016) presented regularized visualizations for maximized activations and inverted images in order to obtain natural looking pre-images.

Using LOTS we explore how the extracted features of an image at any given layer translate back to the input space, and how invariant those features are with respect to the class of a particular input.

### 2.2 Adversarial Representations

Szegedy et al. (2014) revealed that machine learning models, including state-of-the-art DNNs, are vulnerable to adversarial examples that are formed by applying imperceptibly small, non-random perturbations to otherwise correctly recognized inputs leading to misclassifications. This discovery fundamentally challenged our understanding of DNNs, namely, the excellent performance achieved by these complex models was believed to be due to their capability of learning features from the training set that generalize well. The existence of adversarial examples highlights that machine learning models are in fact not robust, and adversarial instability needs to be addressed.

Since Szegedy et al. (2014) presented the problem and introduced the first method that is able to reliably find adversarial perturbations, various approaches were proposed in the literature. Compared to the computationally expensive box-constrained optimization technique (L-BFGS) of Szegedy et al.

(2014), a more lightweight, yet effective technique was introduced by Goodfellow et al. (2015). The proposed fast gradient sign (FGS) method relies on the sign of the gradient of loss which needs to be calculated only once in order to form an adversarial perturbation. The authors also demonstrated that by using FGS examples implicitly in an enhanced objective function, both the overall performance and the robustness of the trained models can be improved.

Both of the aforementioned adversarial example generation techniques rely on ascending the gradient of loss, namely, the formed perturbation causes misclassification by increasing the loss until the particular original class does not have the highest prediction probability. Non-gradient based methods were also proposed by researchers. The approach of Sabour et al. (2016) produces adversarial images that not only cause misclassifications but also mimic the internal representations of the targeted original inputs. Their technique also uses the computationally expensive L-BFGS optimization algorithm. Rozsa et al. (2016) introduced the non-gradient based hot/cold approach, which causes recognition errors by not only reducing the prediction probability of the original class of the input, but by aiming to magnify the probability of the specified targeted class. Therefore, this approach is capable of producing multiple adversarial examples for each input. The authors demonstrated that using a diverse set of such adversarial examples formed with perturbations with higher magnitude than the sufficient minimum necessary to cause misclassifications can outperform regular adversarial training. Finally, Rozsa et al. (2016) proposed a new psychometric called perceptual adversarial similarity score (PASS) to better measure adversarial quality, in other words, the distinguishability or similarity of original and adversarial image pairs in terms of human perception.

Our novel LOTS method can be considered as a general extension of the hot/cold approach to deeper layers, and it also shows similarities to the technique of Sabour et al. (2016) in terms of directly adjusting internal feature representations – without requiring the use of the L-BFGS algorithm.

## 3 APPROACH

One way of gaining insights into the operation of a deep neural network is by letting the network process a given image and – after modifying the output of the network – projecting this modification back to the input level, e.g, via backpropagation. While changing the output appears to be straightforward due to our clear(er) understanding of its meaning, e.g., the hot/cold approach (Rozsa et al., 2016) modifying logits, theoretically, the modification can happen at any layer or any neuron of the network. However, the interpretation of such modifications might be more difficult as the output of a given layer or neuron per se is not guaranteed to have a semantic meaning.

Formally, let us consider a network $f_w$ with weights $w$ in a layered structure, i.e., having layers $y^{(l)}, l = \{1, \ldots, L\}$, with their respective weights $w^{(l)}$. For a given input $x$, the output of the network can be formalized as:

$$f_w(x) = y^{(L)} \left( y^{(L-1)} \left( \ldots \left( y^{(1)}(x) \right) \ldots \right) \right), \tag{1}$$

while the internal representation of the given input $x$ at layer $l$ is:

$$f_w^{(l)}(x) = y^{(l)} \left( y^{(l-1)} \left( \ldots \left( y^{(1)}(x) \right) \ldots \right) \right). \tag{2}$$

Our layerwise origin-target synthesis (LOTS) approach aims to adjust the internal representation of an input $x_o$, the *origin*, to get closer to the internal representation associated with input $x_t$, the *target*. We modify the internal feature representation of $x_o$ at a given layer $l$ to step away from the origin and, in parallel, get closer to the target $x_t$, and project this feature difference back to the input level. To do so, LOTS uses backpropagation operator $B_l$ to estimate input changes on $x_o$ accordingly:

$$\eta^{(l)}(x_o, x_t) = B_l \left( f_w^{(l)}(x_t) - f_w^{(l)}(x_o) \right). \tag{3}$$

We can use the direction defined by the backpropagated feature difference and form adversarial perturbations using a line-search – similar to the fast gradient sign (FGS) method (Goodfellow et al., 2015) and the hot/cold approach (Rozsa et al., 2016). Compared to previous techniques, LOTS has the potential to form dramatically more, and more diverse perturbations for each input due to the large amount of possible targets and the number of layers it can be used on. Also, LOTS is capable of attacking networks that extract deep features rather than doing classification.

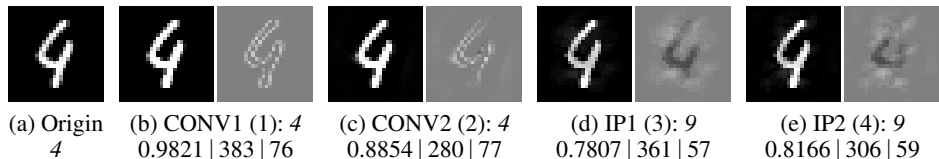

(a) Origin
4

(b) CONV1 (1): *4*
0.9821 | 383 | 76

(c) CONV2 (2): *4*
0.8854 | 280 | 77

(d) IP1 (3): *9*
0.7807 | 361 | 57

(e) IP2 (4): *9*
0.8166 | 306 | 59

Figure 1: VISUALIZATION VIA LOTS ON LENET. *This figure shows perturbed images generated using LOTS by modifying the internal representations of an MNIST image $x_o$, shown in (a), captured at various layers in a trained LeNet model. Each subfigure shows the formed image $x_o^{\pm}$ and the perturbation, while the sub-captions show the name of the layer $l$ followed by an index in parenthesis indicating its position in the network architecture, the classification of $x_o^{\pm}$, the PASS score between origin and the perturbed image, and the $L_2$ and $L_\infty$ norms of the perturbation. (b)–(c) show $x_o^+$ images having magnified internal representations with perturbations scaled such that $L_\infty = 128$. Note that as pixels are constrained to $[0, 255]$, actual $L_\infty$ norms can be smaller. (d)–(e) contain reduced internal representations $x_o^-$ formed by perturbations having the smallest magnitude that changes the class label.*

Alternatively, LOTS can be used by targeting the origin itself, and magnifying or reducing the internal representation of input $x_o$ at layer $l$. This can be formalized as:

$$\hat{\eta}^{(l)}(x_o) = B_l \left( \pm f_w^{(l)}(x_o) \right). \tag{4}$$

While this direction can be utilized to form adversarial perturbations, it can also be used to visualize the captured internal representations of the input at layer $l$. Furthermore, by exploring the magnitudes of perturbations necessary to cause misclassifications, we can assess the robustness of inspected layers with respect to the class of a particular input.

## 4 VISUALIZATION VIA LOTS

First, we focus on demonstrating the visualization capabilities of LOTS, exploring the captured internal representations of inputs on different DNNs. We investigate two publicly available deep neural networks: the 4-layer LeNet network from LeCun et al. (1998) trained on the MNIST dataset (LeCun et al., 1995), and the 16-layer VGG face recognition network from Parkhi et al. (2015).

To quantify the perturbed images, we use three metrics: $L_2$ and $L_\infty$ norms of the perturbations, as well as the perceptual adversarial similarity score (PASS) (Rozsa et al., 2016) between origin $x_o$ and the perturbed image $x_o^{\pm}$. While $L_2$ and $L_\infty$ norms focus strictly on the perturbations regardless of how visible or hidden those are on the distorted images, PASS was designed to quantify the similarity of original and perturbed image pairs with respect to human perception. Therefore, PASS is more suitable to measure adversarial quality with PASS=1 indicating perfect similarity. Note that throughout our experiments, we form perturbed images that have discrete pixel values in $[0, 255]$.

### 4.1 INTERNAL REPRESENTATION OF HANDWRITTEN DIGITS

To visualize the internal representations of layers in LeNet, we apply Equation (4) on each layer. When we use the positive sign to form a perturbed image $x_o^+$, the captured internal representation is magnified and the classification of the modified digit will usually not change. On the other hand, by using the negative sign for forming an image $x_o^-$, we can display which kind of modifications would be required to inhibit the internal representation of origin $x_o$ at that particular layer. Consequently, by increasing the magnitude of the perturbation, the network is more likely to classify $x_o^-$ differently than the origin $x_o$ as feature representations of the original class are fading away.

Figure 1 displays the visualized internal representations of an MNIST image with label 4, shown in Figure 1(a), captured in the LeNet network. The $x_o^+$ samples are displayed in Figures 1(b) and 1(c). Note that magnifying the captured internal feature representations does not lead to altered classifications, therefore, these examples are optimized for visualization purposes. The $x_o^-$ examples for the top layers of LeNet are presented in Figures 1(d) and 1(e), where perturbations have the sufficient minimal magnitude leading to a class label different than the origin's. The perturbations shown in Figures 1(b) and 1(c) demonstrate that the convolutional layers have learned structural characteristics of the input, and increasing the captured internal representations of the origin gracefully translates to a thicker digit. Contrarily, the perturbations formed on the fully-connected layers

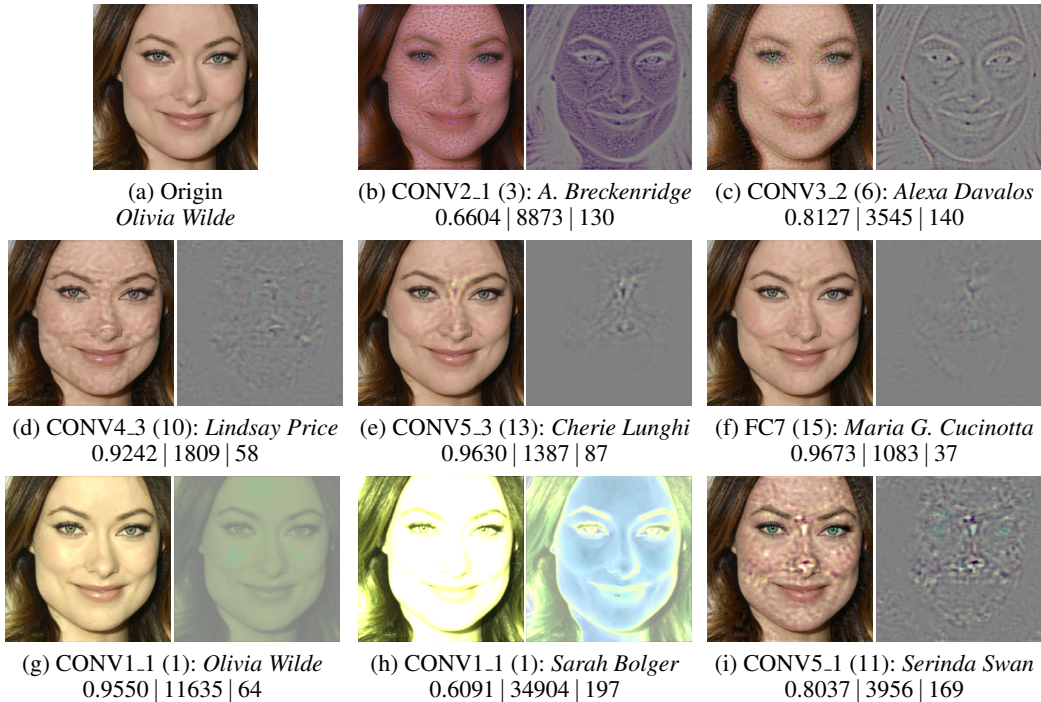

Figure 2: VISUALIZATION VIA LOTS ON VGG FACE. *This figure shows perturbed images generated using LOTS by modifying the internal representations of a face image $x_o$ shown in (a), captured at various layers of the trained VGG Face model. Each subfigure shows a distorted image $x_o^\pm$ and its perturbation. The sub-captions list the name of the layer $l$ followed by an index in parenthesis indicating its position in the network architecture, the classification of $x_o^\pm$, the PASS score between origin and the distorted image, and the $L_2$ and $L_\infty$ norms of the perturbation. (b)–(f) contain reduced internal representations $x_o^-$ with perturbations having the sufficient minimal magnitudes to change the classification. (g) serves purely visualization purposes as its corresponding perturbation is scaled to $L_\infty = 64$. (h)–(i) show $x_o^+$ images that magnify the captured internal representations with sufficiently large perturbations that result in altered classifications by the model.*

displayed in Figures 1(d) and 1(e) seem to have lost spatial focus by considering several locations in the image that appear to be unrelated for the classification of this particular digit.

## 4.2   INTERNAL REPRESENTATION OF FACES

LOTS is capable of displaying more difficult objects, and can be used to visualize more complex classes than the digits of MNIST. To demonstrate this, we generate $x_o^-$ and $x_o^+$ images for various layers of the VGG Face network (Parkhi et al., 2015). Figure 2 shows some distorted samples with the corresponding perturbations generated by modifying the internal representations captured at various layers of the VGG Face network, for an exemplar image taken from the VGG Face Dataset (Parkhi et al., 2015) shown in Figure 2(a). The first two rows of Figure 2 contain $x_o^-$ samples, while the last row shows $x_o^+$ representations.

We can observe that due to the large number of identities (classes) present in the VGG Face Dataset, perturbations lead to various class labels. Also, in opposition to the previously described MNIST samples, both $x_o^-$ and $x_o^+$ representations can yield altered classifications, which means that over-emphasizing the captured features of a given identity also changes the classification of the network, however, these types of samples require perturbations with higher magnitudes.

In summary, the visualized internal feature representations of the origin suggest that lower convolutional layers of the VGG Face model have managed to learn and capture features that provide semantically meaningful and interpretable representations to human observers. Although we can still recognize facial features and parts on visualized feature representations captured at higher layers, closer to the top layer those become harder to interpret. This can be due to the fact that closer to

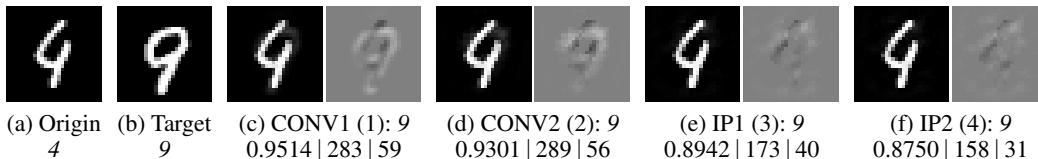

(a) Origin  (b) Target  (c) CONV1 (1): *9*  (d) CONV2 (2): *9*  (e) IP1 (3): *9*  (f) IP2 (4): *9*
*4*  *9*  0.9514 | 283 | 59  0.9301 | 289 | 56  0.8942 | 173 | 40  0.8750 | 158 | 31

Figure 3: ADVERSARIAL PERTURBATIONS VIA LOTS ON LENET. *This figure shows adversarial images and their corresponding perturbations of handwritten digits generated with LOTS using the origin from (a) and target from (b). Each subfigure shows the targeted adversarial image $x_o^t$ and the perturbation, while the sub-captions show the name of layer l followed by an index in parenthesis indicating its position in the network architecture, the classification of $x_o^t$, the PASS score between origin and the adversarial image, and the $L_2$ and $L_\infty$ norms of the perturbation.*

last layer the model needs features that allow for differentiating identities from one another, hence, those features can be small for visualization, and even semantically meaningless for us.

## 5  LOTS OF ADVERSARIAL EXAMPLES

Now, let us exploit the instability of DNNs and show how LOTS can be applied to generate adversarial examples. For a given origin image $x_o$, we aim to form an imperceptibly small perturbation which makes the DNN classify the perturbed image differently than the origin. In order to do so, we can exploit Equation (3) to compute a direction using the feature difference of the origin and target at any given layer, and use it to form the adversarial image such that the perturbation has the lowest magnitude necessary for altering the classification or, if desired, reaching the targeted class.

For adversarial example generation, we use the same networks as before: we commence our experiments forming adversarial perturbations for MNIST digits that exploit the vulnerability of the LeNet model, we compare LOTS with previous techniques, then we turn to the more challenging task of manipulating internal representations of faces to cause misclassifications on the VGG Face model.

### 5.1  ADVERSARIAL EXAMPLES OF HANDWRITTEN DIGITS

Let us consider an image of digit 4 as the origin that we would like to turn to a 9 – at least, when LeNet classifies it. After selecting an applicable image of 9 for being the target, we can use the extracted feature representations of the target and the origin to form the adversarial perturbation. As shown in Figure 3, we use the same origin image as in Figure 1(a) and a selected target image (cf. Figure 3(b)) that looks relatively similar to the origin, but is clearly from another class.

When generating the adversarial example $x_o^t$ by manipulating the internal feature representations captured at the first convolution layer (CONV1), we can clearly see that the perturbation focuses on the difference between the origin 4 and the target 9, as shown in Figure 3(c). Moving to higher layers (CONV2, IP1 and IP2), we can observe that perturbations lose the focus more and more on the global structure. Interestingly, the perturbations have smaller $L_2$ and $L_\infty$ norms, but also lower PASS scores, which indicates that the perturbed images generated at lower layers are less distinguishable from the origin, in other words, they are better in terms of adversarial quality.

In order to compare LOTS with previous adversarial generation techniques, we have conducted experiments with LeNet to measure the adversarial quality of the produced samples and – after using them for training – their effect on error rates and adversarial robustness.

We have trained a LeNet model on the MNIST Training dataset (60K digits), denoted as the Basic LeNet model, and we have also generated 2M images with InfiMNIST (Loosli et al., 2007) and trained a network for 100K iterations with the same hyperparameters distributed with Caffe (Jia et al., 2014). To compare adversarial generation techniques with respect to adversarial quality, we have formed perturbed images with FGS and FGV methods, the hot/cold (HC) approach, and LOTS on the MNIST Test dataset (10K digits), and collected four metrics: the success rate showing the percentage of all attempts the particular technique can produce a perturbation which changes the classification, PASS quantifying the adversarial quality, and $L_2$ and $L_\infty$ norms of perturbations. For LOTS, we have limited the targets by selecting one image per class – yielding "only" 36 possible

Table 1: ADVERSARIAL TRAINING ON LENET. *Collected metrics of adversarial images generated on the MNIST Test dataset using FGS, FGV, HC, and LOTS approaches on corresponding models. The table contains metrics for a Basic LeNet model trained regularly, and for InfiMNIST which is a LeNet model trained with 2M digits. Other models were obtained by generating adversarial examples for the MNIST Training dataset on Basic LeNet and then using those samples to fine-tune Basic LeNet with the adversarial type listed as the model name. For each adversarial type success rate, PASS, and, below, $L_2$ and $L_\infty$ norms are listed.*

| Model | Error | FGS | FGV | HC | LOTS |
|---|---|---|---|---|---|
| Basic LeNet | 0.91% | 85.37% \| 0.4303 ± 0.1098<br>718.8 ± 281.2 \| 34.6 ± 13.2 | 85.37% \| 0.7753 ± 0.1050<br>435.3 ± 187.9 \| 94.4 ± 38.7 | 99.92% \| 0.7269 ± 0.1102<br>596.0 ± 275.5 \| 117.5 ± 47.2 | 99.56% \| 0.6561 ± 0.1245<br>1004.0 ± 456.8 \| 170.0 ± 52.4 |
| InfiMNIST | 0.53% | 92.89% \| 0.4553 ± 0.1069<br>729.6 ± 260.6 \| 35.2 ± 12.5 | 92.88% \| 0.7819 ± 0.0921<br>450.5 ± 181.7 \| 94.6 ± 36.2 | 99.71% \| 0.7417 ± 0.0944<br>566.0 ± 224.9 \| 107.6 ± 40.4 | 99.07% \| 0.6667 ± 0.1236<br>986.8 ± 473.7 \| 163.6 ± 54.1 |
| FGS | 0.77% | 80.69% \| 0.4835 ± 0.1019<br>650.9 ± 255.8 \| 33.6 ± 13.8 | 80.69% \| 0.7576 ± 0.0997<br>427.7 ± 186.2 \| 91.0 ± 39.7 | 99.76% \| 0.7200 ± 0.1022<br>594.3 ± 249.0 \| 116.8 ± 49.8 | 91.05% \| 0.6457 ± 0.1138<br>1032.0 ± 393.4 \| 179.6 ± 51.5 |
| FGV | 0.82% | 60.52% \| 0.4240 ± 0.1161<br>701.3 ± 256.0 \| 35.2 ± 13.2 | 60.52% \| 0.7722 ± 0.1083<br>434.5 ± 189.8 \| 96.1 ± 39.1 | 99.95% \| 0.7140 ± 0.1079<br>684.8 ± 311.2 \| 135.9 ± 52.4 | 99.69% \| 0.6361 ± 0.1242<br>1094.3 ± 431.3 \| 188.8 ± 50.8 |
| HC | 0.78% | 71.92% \| 0.4527 ± 0.1116<br>683.3 ± 263.8 \| 34.5 ± 13.7 | 71.92% \| 0.7669 ± 0.1105<br>449.8 ± 231.7 \| 95.0 ± 42.7 | 99.99% \| 0.7140 ± 0.1064<br>653.1 ± 320.5 \| 126.6 ± 53.2 | 99.89% \| 0.6401 ± 0.1213<br>1074.0 ± 414.0 \| 183.8 ± 51.9 |
| LOTS | 0.65% | 30.89% \| 0.4506 ± 0.1250<br>597.5 ± 220.1 \| 29.4 ± 10.9 | 30.88% \| 0.8208 ± 0.1044<br>341.1 ± 132.9 \| 83.6 ± 32.7 | 97.26% \| 0.7004 ± 0.1127<br>686.1 ± 271.8 \| 143.1 ± 50.3 | 97.61% \| 0.6285 ± 0.1239<br>1119.6 ± 422.9 \| 194.0 ± 48.2 |
| LOTS CONV1 | 1.02% | 47.58% \| 0.4783 ± 0.1316<br>486.7 ± 185.4 \| 23.9 ± 9.1 | 47.58% \| 0.8144 ± 0.1030<br>295.3 ± 117.5 \| 66.2 ± 26.7 | 100.00% \| 0.7374 ± 0.1025<br>538.5 ± 231.5 \| 106.4 ± 41.7 | 90.03% \| 0.6471 ± 0.1357<br>1036.8 ± 538.8 \| 171.6 ± 55.0 |
| LOTS CONV2 | 0.90% | 54.86% \| 0.4679 ± 0.1263<br>533.6 ± 192.5 \| 26.4 ± 9.5 | 54.85% \| 0.8169 ± 0.1047<br>317.6 ± 120.1 \| 73.3 ± 28.4 | 99.92% \| 0.7468 ± 0.1046<br>535.9 ± 216.1 \| 110.2 ± 41.6 | 98.26% \| 0.6518 ± 0.1328<br>1044.2 ± 500.6 \| 177.9 ± 53.5 |
| LOTS IP1 | 0.74% | 46.16% \| 0.4456 ± 0.1257<br>659.1 ± 238.9 \| 32.5 ± 11.9 | 46.16% \| 0.8049 ± 0.1131<br>386.5 ± 159.5 \| 90.9 ± 36.2 | 99.33% \| 0.7137 ± 0.1204<br>662.2 ± 256.8 \| 136.2 ± 50.0 | 98.72% \| 0.6319 ± 0.1261<br>1099.4 ± 416.2 \| 189.7 ± 48.9 |
| LOTS IP2 | 0.78% | 46.43% \| 0.4263 ± 0.1275<br>717.8 ± 266.8 \| 35.6 ± 13.6 | 46.42% \| 0.7724 ± 0.1209<br>427.5 ± 172.1 \| 96.3 ± 37.5 | 99.39% \| 0.6869 ± 0.1155<br>727.8 ± 292.7 \| 144.7 ± 53.5 | 99.17% \| 0.6239 ± 0.1215<br>1110.7 ± 387.2 \| 191.8 ± 48.9 |

perturbations for each input. Compared to other techniques, LOTS maintains a high success rate while the achieved adversarial quality is lower than FGV and HC samples have, as we can see in Table 1. However, we have found that using LOTS with FGV or HC targets can produce perturbed images with adversarial quality surpassing the originating targets – indicated by higher PASS scores – but these "improved" samples do not have any further benefits when used for adversarial training.

To analyze the performances of adversarial generation techniques when the formed samples are used for training, we have generated perturbed images for the MNIST Training dataset on our Basic LeNet model and then used those images for fine-tuning their originating learning model. Since we have various numbers of perturbed images for each adversarial type, we have fine-tuned Basic LeNet using the regular hyperparameters for different number of iterations: 20K iterations for FGS (app. 52K images) and FGV (app. 51K images) samples, 50K iterations for HC (app. 538K images), and 100K iterations for LOTS (app. 2.1M images) samples. For LOTS, we have selected one image per class and used the same 10 images as targets. Considering all possible targets, the overall number of perturbed images would be beyond 10 billion. By looking at the results listed in Table 1, we can observe that fine-tuning with LOTS samples achieves the lowest error rate (0.65%) on the MNIST Test dataset among models trained with adversarial examples – slightly worse than InfiMNIST – and this network is the most robust considering the collected metrics of all four adversarial types as well. Finally, we have compared LOTS samples formed by manipulating the captured feature representations at various layers. We have found that, in general, fine-tuning with LOTS samples from higher layers require adversarial generation techniques to form stronger, more visible perturbations as indicated by the decreasing PASS scores and increasing $L_2$ and $L_\infty$ norms on models denoted as LOTS CONV1, LOTS CONV2, LOTS IP1, and LOTS IP2.

## 5.2 ADVERSARIAL EXAMPLES OF FACES

A greater threat to automatic face recognition systems is presented by the possibility of generating adversarial examples from face images. In DNN-based face recognition systems, usually the last layer of the network (containing the identities) is disregarded, and the output of the penultimate layer (e.g., FC7 of the VGG Face model) is stored as a representation of the face. When an adversary steals this representation of a target (without requiring the possession of the target image) and also has access to the original network, he can generate an image that looks like himself to a human operator, but is identified as the target identity by the network.

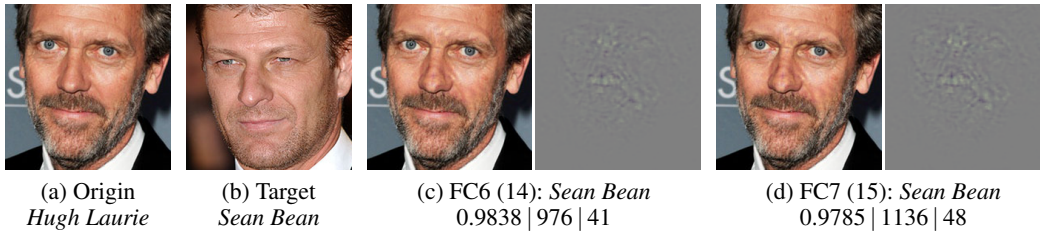

| (a) Origin | (b) Target | (c) FC6 (14): *Sean Bean* | (d) FC7 (15): *Sean Bean* |
|:---:|:---:|:---:|:---:|
| *Hugh Laurie* | *Sean Bean* | 0.9838 \| 976 \| 41 | 0.9785 \| 1136 \| 48 |

Figure 4: ADVERSARIAL PERTURBATIONS VIA LOTS ON VGG FACE. *This figure shows adversarial images and their corresponding perturbations of face images generated with LOTS using the origin from (a) and target from (b). Each subfigure shows the targeted adversarial image $x_o^t$ and the perturbation, while the sub-captions show the name of the layer $l$ followed by an index in parenthesis indicating its position in the network architecture, the classification of $x_o^t$, the PASS score between origin and the adversarial example, and the $L_2$ and $L_\infty$ norms of the perturbation.*

Given the internal representation of a target image $x_t$ representing identity $t$ (*Sean Bean* in Figure 4(b)) at a given layer $l$ of the network, and an origin image $x_o$ including its internal representation of identity $o$ (*Hugh Laurie* in Figure 4(a)), we can use Equation (3), and form the adversarial image $x_o^t$ classified as identity $t$. As shown in Figure 4, the adversarial images are basically indistinguishable from the original images as indicated by the very high PASS scores, yet the network classifies them incorrectly as the target.

Finally, LOTS can be modified to perform better, however, it would be computationally more expensive. Namely, instead of taking the direction defined by the backpropagated feature difference with respect to the origin once and using it throughout the whole line-search, we can consider calculating the direction multiple times. This "step-and-adjust" optimization could potentially improve both the quality of the produced adversarial examples and the capability of LOTS to reach the target.

## 6 CONCLUSION

In this paper, we have presented our novel layerwise origin-target synthesis (LOTS) algorithm which can be efficiently used for multiple purposes as we have demonstrated on two well-known deep neural networks (DNNs): the hand-written digit recognition network LeNet and the face recognition network from VGG. First, we can visualize the captured internal structure of an input at any layer of DNNs and we can also gain intuitive insights into the interpretation of a given input by DNNs by magnifying what the network at a particular layer "thinks" is or is not a good representation of the object. Second, the stability of the captured internal feature representations can be assessed with respect to the class of the input by exploring how robust they are to perturbations aiming at magnifying or reducing them. Third, we have demonstrated that LOTS is capable of producing a large number of diverse adversarial examples for each input by modifying the internal structure of the original input to mimic the particular target. This application of LOTS can help us get a deeper understanding about the intrinsic features that differentiate classes in various layers of DNNs. Finally, we have conducted large-scale experiments to compare our approach with other adversarial generation techniques and have concluded that using the large number of diverse perturbed examples produced via LOTS for adversarial training outperforms previous methods with respect to both the achieved performance and the improved adversarial robustness.

ACKNOWLEDGMENTS

This research is based upon work funded in part by NSF IIS-1320956 and in part by the Office of the Director of National Intelligence (ODNI), Intelligence Advanced Research Projects Activity (IARPA), via IARPA R&D Contract No. 2014-14071600012. The views and conclusions contained herein are those of the authors and should not be interpreted as necessarily representing the official policies or endorsements, either expressed or implied, of the ODNI, IARPA, or the U.S. Government. The U.S. Government is authorized to reproduce and distribute reprints for Governmental purposes notwithstanding any copyright annotation thereon.

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
