# Peer review of "Exploring LOTS in Deep Neural Networks"

_ICLR 2017 — rejected_

[Official Review · AnonReviewer1 · rating 6 · confidence 4 · 14 Dec 2016 (modified: 23 Jan 2017)]
**Interesting but somewhat incomplete analysis**

This paper presents a relatively novel way to visualize the features / hidden units of a neural network and generate adversarial examples. The idea is to do gradient descent in the pixel space, from a given hidden unit in any layer. This can either be done by choosing a pair of images and using the difference in activations of the unit as the thing to do gradient descent over or just the activation itself of the unit for a given image. In general this method seems intriguing, here are some comments:

It’s not clear that some of the statements at the beginning of Sec 4.1 are actually true, re: positive/negative signs and how that changes (or does not change) the class. Mathematically, I don’t see why that would be the case? Moreover the contradictory evidence from MNIST vs. faces supports my intuition.

The authors use the PASS score through the paper, but only given an intuition + citation for it. I think it’s worth explaining what it actually does, in a sentence or two.

The PASS score seems to have some, but not complete, correlation with L_2, L_\{infty} or visual estimation of how “good” the adversarial examples are. I am not sure what the take-home message from all these numbers is.

“In general, LOTS cannot produce high quality adversarial examples at the lower layers” (sec 5.2) seems false for MNIST, no?

I would have liked this work to include more quantitative results (e.g., extract adversarial examples at different layers, add them to the training set, train networks, compare on test set), in addition to the visualizations present. That to me is the main drawback of the paper, in addition to basically no comparisons with other methods (it’s hard to judge the merits of this work in vacuum).

-----

EDIT after rebuttal: thanks to the authors for addressing the experimental validation concerns. I think this makes the paper more interesting, so revising my score accordingly.

[Official Review · AnonReviewer3 · rating 6 · confidence 4 · 16 Dec 2016]
**Exciting new method to generate adversarial examples and study robustness, less interesting analyses**

The paper presents a new exciting layerwise origin-target synthesis method both for generating a large number of diverse adversarials as well as for understanding the robustness of various layers. The methodology is then used to visualize the amount of perturbation necessary for producing a change for higher level features.

The approach to match the features of another unrelated image is interesting and it goes beyond producing adversarials for classification. It can also generate adversarials for face-recognition and other models where the result is matched with some instance from a database.

Pro: The presented approach is definitely sound, interesting and original. 
Con: The analyses presented in this paper are relatively shallow and don't touch the most obvious questions. There is not much experimental quantitative evidence for the efficacy of this method compared with other approaches to produce adversarials. The visualization is not very exciting and it is hard to any draw any meaningful conclusions from them.

It would definitely improve the paper if it would present some interesting conclusions based on the new ideas.

[Public Comment · Alexey Kurakin · 26 Dec 2016]
**Question about eq 3 and 4**

Hi,

I have a question about eq. 3 and 4.

As far as I understood from the beginning of section 3, f_{w}^{l} (x) is essentially activations at layer l, which means that value of f_{w}^{l} (x) is in R^{d_l} space where d_l is dimensionality of output of layer l.
In eq. (3) and (4) you \eta compute gradient of f_{w}^{l} (x) over input x, which I would expect to be Jacobian matrix with size d_l*n (where n - dimensionality of input x). So looking at eq (3) and (4) I would expect that \eta and x has different dimensionality. At the same time in section 4 you add s*\eta to x.
Could you explain this discrepancy in dimensionality and how \eta should be calculated?
Maybe you meant that numerator of eq (3) and (4) contain norm of f_{w}^{l} instead of it's value?

Thanks,
Alex

[Official Review · AnonReviewer2 · rating 6 · confidence 4 · 27 Dec 2016 (modified: 20 Jan 2017)]
**Great start; recommended as workshop paper.**

This paper proposes the Layerwise Origin Target Synthesis (LOTS) method, which entails computing a difference in representation at a given layer in a neural network and then projecting that difference back to input space using backprop. Two types of differences are explored: linear scalings of a single input’s representation and difference vectors between representations of two inputs, where the inputs are of different classes.

In the former case, the LOTS method is used as a visualization of the representation of a specific input example, showing what it would mean, in input space, for the feature representation to be supressed or magnified. While it’s an interesting computation to perform, the value of the visualizations is not very clear.

In the latter case, LOTS is used to generate adversarial examples, moving from an origin image just far enough toward a target image to cause the classification to flip. As expected, the changes required are smaller when LOTS targets a higher layer (in the limit of targetting the last layer, results similar to the original adversarial image results would be obtained).

The paper is an interesting basic exploration and would probably be a great workshop paper. However, the results are probably not quite compelling enough to warrant a full ICLR paper.

A few suggestions for improvement:
 - Several times it is claimed that LOTS can be used as a method for mining for diverse adversarial examples that could be used in training classifiers more robust to adversarial perturbation. But this simple experiment of training on LOTS generated examples isn’t tried. Showing whether the LOTS method outperforms, say, FGS would go a long way toward making a strong paper.
 - How many layers are in the networks used in the paper, and what is their internal structure? This isn’t stated anywhere. I was left wondering whether, say, in Fig 2 the CONV2_1 layer was immediately after the CONV1_1 layer and whether the FC8 layer was the last layer in the network.
 - In Fig 1, 2, 3, and 4, results of the application of LOTS are shown for many intermediate layers but miss for some reason applying it to the input (data) layer and the output/classification (softmax) layer. Showing the full range of possible results would reinforce the interpreatation (for example, in Fig 3, are even larger perturbations necessary in pixel space vs CONV1 space? And does operating directly in softmax space result in smaller perturbations than IP2?)
 - The PASS score is mentioned a couple times but never explained at all. E.g. Fig 1 makes use of it but does not specify such basics as whether higher or lower PASS scores are associated with more or less severe perturbations. A basic explanation would be great.
 - 4.2 states “In summary, the visualized internal feature representations of the origin suggest that lower convolutional layers of the VGG Face model have managed to learn and capture features that provide semantically meaningful and interpretable representations to human observers.” I don’t see that this follows from any results. If this is an important claim to the paper, it should be backed up by additional arguments or results.



1/19/17 UPDATE AFTER REBUTTAL:
Given that experiments were added to the latest version of the paper, I'm increasing my review from 5 -> 6. I think the paper is now just on the accept side of the threshold.

[Author Response · Andras Rozsa · 07 Jan 2017]
**Revision addressing comments and suggestions**

Our revised paper reflects to comments and suggestions for improvement.

(1) We have added quantitative results of large-scale experiments comparing our LOTS approach to other adversarial example generation techniques such as the fast gradient sign (FGS) method, fast gradient value (FGV) method, and hot/cold (HC) approach. We have analyzed these techniques with respect to the quality of the produced images, and have also evaluated the effect of those samples when they are used for adversarial training.
(2) We have clarified the motivation and advantage of using PASS over L-2 and L-inf norms with respect to measuring adversarial quality.
(3) We have provided details about the internal structure of the tested network architectures, e.g., by denoting the position of layers in figures.
(4) We have reviewed and simplified the whole paper to enhance readability and, last but not least, to allow us presenting the new results.

Thank you for all your comments and suggestions!

[Final Decision · Program Chairs · 06 Feb 2017]
**ICLR committee final decision**

This paper studies the effects of modifying intermediate representations arising in deep convolutional networks, with the purpose of visualizing the role of specific neurons, and also to construct adversarial examples. The paper presents experiments on MNIST as well as faces. 
 
 The reviewers agreed that, while this contribution presents an interesting framework, it lacks comparisons with existing methods, and the description of the method lacks sufficient rigor. In light of the discussions and the current state of the submission, the AC recommends rejection. 
 
 Since the final scores of the reviewers might suggest otherwise, please let me explain my recommendation. 
 
 The main contribution of this paper seems to be essentially a fast alternative to the method proposed in 'Adversarial Manipulation of Deep Representations', by Sabour et al, ICLR'16, although the lack of rigor and clarity in the presentation of section 3 makes this assessment uncertain. The most likely 'interpretation' of Eq (3) suggests that eta(x_o, x_t) = nabla_{x_o}( || f^(l)_w(x_t) - f^(l)_w(x_o) ||^2), which is simply one step of gradient descent of the method described in Sabour et al. One reviewer actually asked for clarification on this point on Dec. 26th, but the problem seems to be still present in the current manuscript. 
 
 More generally, visualization and adversarial methods based on backpropagation of some form of distance measured in feature space towards the pixel space are not new; they can be traced back to Simoncelli & Portilla '99. 
 Fast approximations based on simply stopping the gradient descent after one iteration do not constitute enough novelty. 
 
 Another instance of lack of clarity that has also been pointed out in this discussion but apparently not addressed in the final version is the so-called PASS measure. It is not defined anywhere in the text, and the authors should not expect the reader to know its definition beforehand. 
 
 Besides these issues, the paper does not contribute to the state-of-the-art of adversarial training nor feature visualization, mostly because its experiments are limited to mnist and face datasets. Since the main contribution of the paper is empirical, more emphasis should be made to present experiments on larger, more numerous datasets.